# Effects of Organic Based Heat Stabilizer on Properties of Polyvinyl Chloride for Pipe Applications: A Comparative Study with Pb and CaZn Systems

**DOI:** 10.3390/polym14010133

**Published:** 2021-12-30

**Authors:** Chanchira Jubsilp, Aran Asawakosinchai, Phattarin Mora, Duangporn Saramas, Sarawut Rimdusit

**Affiliations:** 1Department of Chemical Engineering, Faculty of Engineering, Srinakharinwirot University, Nakhon Nayok 26120, Thailand; chanchira@g.swu.ac.th (C.J.); phattarin@g.swu.ac.th (P.M.); 2Research Unit in Polymeric Materials for Medical Practice Devices, Department of Chemical Engineering, Faculty of Engineering, Chulalongkorn University, Pathumwan, Bangkok 10330, Thailand; llxartxll@hotmail.com; 3Vinythai PLC, 2 Map Ta Phut Industrial Estate, I-3 Rd., Map Ta Phut, Muang, Rayong 21150, Thailand; Duangporn.sara@agc.com

**Keywords:** fusion time, gelation, color change, activation energy, mechanical property

## Abstract

In this paper, the effects of organic based stabilizers (OBS) are investigated and compared with traditional lead (Pb) and calcium zinc (CaZn) heat stabilizers regarding their processability, mechanical property, and thermal degradation behaviors in rigid PVC pipe applications. In addition, the effects of repeated processing cycles on the degree of gelation and the impact strength of the PVC/OBS, PVC/CaZn, and PVC/Pb are also examined. A repeated processing cycle of those three types of the heat stabilizers up to four cycles was found to increase the degree of gelation and proved no significant effect on the impact strength and heat resistance of the resulting PVC samples. The OBS showed a positive effect on preventing the autocatalytic-typed thermal degradation of the PVC samples. This leads to a longer retention time for the initial color change of the PVC/OBS compared to PVC/Pb or PVC/CaZn systems. This characteristic was related to a more uniform fusion behavior of the PVC/OBS, i.e., the lowest gelation speed and the longest fusion time. The non-isothermal kinetic parameter determined by the Kissinger and Flynn–Wall–Ozawa methods of the dehydrochlorination stage of the PVC/OBS was in satisfactory agreement and continued to compare with the PVC/Pb and PVC/CaZn systems. The results indicated that the OBS might decrease the dehydrochlorination rate of PVC, implying that PVC/OBS was more stable than PVC/Pb and PVC/CaZn systems.

## 1. Introduction

Polyvinyl chloride (PVC) is a well-known commodity plastic with a broad spectrum of properties that are rigid, semi-flexible, and flexible. It is widely used in various construction applications, including water, sewage, or drainage pipes and in many extruded profiles. The rigid PVC form has broad acceptance and exhibits a good market growth in major PVC product applications, i.e., pipes, due to its outstanding functional properties and cost competitiveness [1,2]. However, PVC is thermally unstable at processing temperatures. When PVC is heated to 170–180 °C, chlorine and hydrogen in the molecules are eliminated, and it releases hydrochloric acid (HCl), which in turn accelerates the thermal degradation process of the number of conjugated double bonds formed during the processing. It also causes the level of color in the sample to range from yellow to black [1,2,3]. Hydrochloric acid can also deteriorate the mechanical, thermal, and physical properties of PVC. Therefore, heat stabilizers were widely used to safeguard the vinyl products at all stages by improving the heat resistance of PVC products at high temperatures, preventing the chain reaction of degradation. They can also impart PVC to enhance daylight resistance, weathering, and heat ageing. Recently, heat stabilizers contributed more than 30% of the global PVC additive demand. Heavy metal-based heat stabilizers such as lead (Pb)-based products are responsible for a large proportion of products in the heat stabilizer group. These products are widely used for the stabilization of pipe in Europe and Asia (except Japan). However, in 2003, the European Union passed regulations that stated the use of Pb will be limited in the production of PVC pipes, as well as in a number of other manufacturing processes. As a consequence, considerable efforts have been expended towards the development of new stabilization systems for PVC processing, which is driven by a desire to move away from stabilizers based on heavy metals [4,5]. An alternative stabilizer to replace Pb-based stabilizers such as calcium zinc (CaZn) is investigated. CaZn stabilizers are preferred for use in critical applications because of low toxicity; however, they are the least effective option in terms of thermal performance. In addition, CaZn was used by pipe producers but not advanced further due to relatively high costs [5]. Therefore, to ensure that PVC meets the regulations and remains an acceptable material for use in pipe applications, organic-based stabilizers (OBSs) have been developed to replace traditional Pb-based stabilizers and to overcome the disadvantage in long-term stability of CaZn-based stabilizers.

Organic based stabilizers for PVC have been recognized as a new technology for providing environmentally friendly heat stabilizers in PVC pipe production. Their usage has been increased in recent years due to economic growth in emerging markets and increased knowledge about the benefits of organic based stabilizers for PVC [6,7]. In addition, organic based stabilizers deliver several other advantages, such as outstanding paint fastness even after many processing rounds. In fact, it typically consists of components that are stated positively, making it ideal for interaction with food. Organic based stabilizers for PVC exhibit high thermal stability and excellent properties for recycling. These reasons have motivated producers all over the world, and many are switching to organic alternatives because of legislation, market pressure, or on a voluntary basis. However, there is very little, or no information in comparison to rheology, thermal stability in terms of color change, heat resistance, and thermal degradation kinetic as well as the recycling ability of PVC stabilized with traditional Pb and CaZn heat stabilizers. There is also a lack of comparison to OBS which is a more eco-friendly heat stabilizer, essentially free of heavy metals—such as lead, barium, zinc, tin, or cadmium—which contains calcium in these solid metals. Therefore, in this present work, we are motivated to investigate the performance of organic based stabilizers (OBS). The rheological behaviors, color change, thermal stability, and mechanical property of PVC/OBS systems compared with those of PVC/Pb and PVC/CaZn systems were also studied. The recycling ability of the PVC/OBS, PVC/CaZn, and PVC/Pb was evaluated. PVC/OBS was expected to have a better performance than PVC/CaZn and PVC/Pb. This work also focused on thermal degradation kinetics through TGA measurement under non-isothermal conditions in a nitrogen atmosphere for the PVC/OBS, which, to the best of our knowledge, has not been reported on in the literature. The kinetic parameters based on the Kissinger method, using the peak temperature as the maximum of the reaction rate, and the isoconversional method, Flynn–Wall–Ozawa (FWO), based on the extent of reaction (conversion), were studied.

## 2. Materials and Methods

### 2.1. Materials

Suspension-grade PVC resin with a K-value of 66, calcium carbonate filler, and a lead (Pb) stabilizer was supplied by Vinythai PCL (Bangkok, Thailand). Calcium zinc (CaZn) stabilizer was supplied by Dalian Chemson Technical Co., Ltd. (Dalian, China). Organic based stabilizer (OBS) was provided by Brenntag Ingredients (Thailand) PCL (Bangkok, Thailand). All PVC heat stabilizers, i.e., Pb, CaZn, and OBS are one-pack systems.

### 2.2. Sample Preparation

PVC compounds were prepared from PVC resin, calcium carbonate, and heat stabilizers such as Pb, CaZn, and OBS at a mass content of 3 phr. The PVC compounds were mixed in a high-speed mixer (Plas Mec Turbomixer, Varese-Italy). The mixing process consists of two mixing tanks, operating with two steps of hot and cold blending using temperature levels of 70 and 30 °C, respectively. The PVC compounds were then processed by two-roll mills with a 0.125-mm gap at a temperature of 180 °C for 3 min to yield a homogeneous sample in sheet form.

### 2.3. Research Methods

#### 2.3.1. Physical Property

The fusion behaviors of the PVC compounds were measured by torque rheometer (Brabender kneader, Plasti–Corder Pl, Duisburg-Germany). The chamber temperature was kept at 180 °C, and the screw rotational speed was approximately 60 rpm.

Discoloration of strips was obtained by two-roll mills with a sample size of 1.5 cm × 30 cm × 1.0 mm and was observed in static thermal stability tests. The strips were put in the Mathis thermotester (Zurich, Switzerland) at a temperature of 180 °C and held for 5 min and then automatically moved out of the box at 1 mm/min.

The degree of gelation was measured using differential scanning calorimetry (DSC), model DSC1, STARe system, Mettler-Toledo (Zurich, Switzerland). Each sample of 10 mg was put in an aluminum pan with a lid and scanned at a heating rate of 10 °C/min from 50 to 250 °C under nitrogen purging with a constant flow of 50 mL/min.

#### 2.3.2. Mechanical Property

Flexural properties of samples were tested using a universal testing machine (UTM), model 5567, Instron (Massachusetts, USA), according to ASTM D790. Three-point bending was carried out using a crosshead speed of 1.2 mm/min with a support span of 48 mm. The sample dimension was 4 mm × 12.7 mm × 64 mm. A five-averaged value was taken and reported.

The notched Izod impact strength of the samples was measured using an impact tester (Yasuda, Chiba-Japan) according to ASTM D256. The sample dimension was 12.7 mm × 60 mm × 3.2 mm, and the depth under the notch of the sample was 10.7 mm.

#### 2.3.3. Thermal Degradation Analysis

TGA thermograms of samples were acquired using thermogravimetric analysis (TGA) (model TGA1 STARe system, Mettler-Toledo, Zurich-Switzerland). A sample of approximately 10 mg was used for each test. The temperature program was heated with a heating rate of 20 °C/min from 25 to 800 °C, under nitrogen purging with a constant flow of 50 mL/min.

The thermal degradation kinetics of the samples were studied by a thermogravimetric analyzer. Each sample was scanned from room temperature to 800 °C under the nitrogen atmosphere at heating rates of 5, 10, and 20 °C/min. Kinetic parameters, such as activation energy (*E*_a_, kJ/mol), were calculated using the Kissinger and Flynn–Wall–Ozawa (FWO) methods.

#### 2.3.4. Kinetic Analysis and Methods

According to the non-isothermal kinetic theory of solid-state reactions [8,9], the rate of conversion of the sample (d*α*/d*t*) is proportional to the concentration of the reacted sample. It is a linear function of a temperature-dependent rate constant, *k*, and a temperature-independent function of conversion (*α*), that is:(1)$$\frac{d\alpha }{dt}=kf\left(\alpha \right)\;$$
(2)$$\alpha =\frac{m_{0}-m_{t}}{m_{0}-m_{f}}\;$$
where *m_t_* is the measured experimental mass at time *t*, *m*_0_ is the initial mass, and *m_f_* is the mass at the end of the non-isothermal experiments.

Substituting the Arrhenius equation $k=Aexp\left(\frac{-E_{a}}{RT}\right)$ into Equation (1), obtains:(3)$$\frac{d\alpha }{dt}=Aexp\left(\frac{-E_{a}}{RT}\right)f\left(\alpha \right)$$
where *A* is the Arrhenius pre-exponential factor (1/min), *E_a_* is the activation energy (kJ/mol), *T* is the absolute temperature (K), *R* is the gas constant (8.314 J/mol K), and *f*(*α*) accounts for the reaction rate dependence on *α*. The kinetic model, *f*(*α*), is an algebraic expression that is usually associated with a physical model that describes the kinetics of the solid-state reaction.

For non-isothermal kinetic theory, the temperature of the sample is changed by a controlled and constant heating rate, *β*(K/min) = *dT*/*dt*. Therefore, the reaction rate is given by:(4)$$\frac{d\alpha }{dT}=\frac{A}{\beta }exp\left(\frac{-E_{a}}{RT}\right)f\left(\alpha \right)\;$$

Integration of this equation from an initial temperature (*T*_0_) corresponding to a null degree of conversion, to the peak temperature of the derivative thermogravimetric curve (DTG), *T*_p_, where *α* = *α_p_* gives:(5)$$g\left(\alpha \right)=\int_{0}^{\alpha_{p}}\frac{d\alpha }{f\left(\alpha \right)}=\frac{A}{\beta }\int_{0}^{\alpha_{p}}exp\left(\frac{-E_{a}}{RT}\right)dT\;$$
where g(α) is the integral function of conversion.

Kissinger method (Differential method) [10,11,12];

The Kissinger method is used to determine the activation energy of solid-state reactions. From the slope of the plots between the logarithms of the heating rate and the inverse of temperature at the maximum of the reaction rate in constant heating rate experiments, the activation energy can be obtained without precise knowledge of any thermal degradation reaction mechanism, using the following equation:(6)$$ln\frac{\beta }{T_{p}^{2}}=\left\{ln\frac{AR}{E_{a}}+ln\left[n{\left(1-\alpha_{p}\right)}^{n-1}\right]\right\}-\frac{E_{a}}{RT_{p}}\;$$
where *E_a_* is the activation energy, *β* is the heating rate; *T_p_* is the temperature corresponding to the inflection point of the thermal degradation curves, which corresponds to the maximum reaction rate; *A* is the pre-exponential factor; *α_p_* is the maximum conversion; and *n* is the reaction order.

2Flynn–Wall–Ozawa (FWO) method (Isoconversional integral method) [13,14]

FWO is one of the integral methods used to calculate the kinetic parameters using Doyle’s approximation of a temperature integral. This method can be employed to quantify the activation energy at given values of conversion (*E_a,_**_∝_*) without any knowledge of the reaction mechanisms, using the following equation:(7)$$log\beta =log\left[\frac{A_{\alpha }E_{a,\alpha }}{g\left(\alpha \right)R}\right]-2.315-\frac{0.456E_{a,\alpha }}{RT_{\alpha }}\;$$
where $g\left(\alpha \right)=\int_{0}^{\alpha }\frac{d\alpha }{f\left(\alpha \right)}$ is the integral conversion function.

## 3. Results

### 3.1. Fusion Characteristics of PVC Compounds Stabilized with Various Heat Stabilizers

Fusion characteristics, consisting of fusion torque, fusion time, and fusion temperature, are an important rheological behavior that helps to understand the processability of PVC compounds and machine safety. The processability of PVC compounds is usually measured in terms of fusion characteristics. It is a well-known fact that a grain of PVC compounds is the largest unit of primary particles having an agglomeration, approximately 1 µm. During the processing of PVC grains, primary particles are broken down to produce a continuous phase. Therefore, the rheological data, such as torque and temperature at constant rotational speed versus time of PVC compounds under processing conditions, was measured. The plastograms of the fusion characteristics measured from the torque rheometer of PVC compounds, i.e., PVC/Pb, PVC/CaZn, and PVC/OBS, are presented in Figure 1. From the plots, the red line was a real PVC compound temperature, whereas the blue line was the mixing torque as a function of mixing time. After the loading stage, a significant increase of the torque was observed (point A), while the temperature also rose due to frictional heat. The torque started to decrease near point A until point B mainly due to the onset of homogenization and rose in temperature as a result of shear. At point B, the torque tended to increase because of the compaction, melting, and entanglement of the PVC molecules at the primary particle surface. Then, the inflection was observed at point G, at which the PVC compound was melted and the torque rose again up to X. Thereafter, cross-linking was facilitated. For the temperature curve, the temperature increased gradually, leading to the fusion at point G. Finally, the temperature reached a steady-state after point X. After point X, a slight decrease and stabilization of the torque can be observed at point E, whereas the temperature between points X and E remained relatively constant [15].

The rheological data, such as torque and temperature at a constant rotational speed versus time of the PVC/Pb, PVC/CaZn, and PVC/OBS compounds, can be seen in Figure 1 and tabulated in Table 1. It could be observed that the fusion time of the PVC/Pb and PVC/CaZn is shorter than that of the PVC/OBS. The order of fusion time is PVC/OBS > PVC/CaZn > PVC/Pb. This may be due to the fact that the PVC/Pb and PVC/CaZn interacted and formed powerful elements with less absorbed energy compared to PVC/OBS. Therefore, the PVC/Pb and PVC/CaZn tends to be easier to process than the PVC/OBS. However, the degradation behavior of the compounds is also considered in PVC processing. From the fusion time values, it is indicated that the PVC/OBS would degrade slower than the PVC/Pb and the PVC/CaZn. This result was in agreement with the longer degradation time (the difference in time between A and E) of PVC/OBS measured as 895 s compared to that of the PVC/Pb and the PVC/CaZn along 590 s and 892 s, respectively. In addition, the degradation behavior of the PVC compounds can also be evaluated from the gelation speed related to the crosslinking of the compounds under processing conditions. From Table 1, the lowest gelation speed related to the highest crosslinking belonged to the PVC/OBS. It confirms that the PVC/OBS compound was degraded slower than the other two aforementioned compounds. Therefore, it summarizes that the addition of OBS may significantly inhibit the degradation behavior of the PVC compound compared to the addition of Pb and Ca/Zn stabilizers.

### 3.2. Static Thermal Stability of PVC Stabilized with Various Heat Stabilizers

Static thermal stability of rigid PVC/Pb, PVC/CaZn, and PVC/OBS samples processed by two roll mills at 180 °C for 3 min was revealed by the observation of a visual change in color due to heating at 180 °C as a function of time, as shown in Figure 2. However, this practice is particularly applicable for determining gross differences in the heat stabilities of the PVC compositions that are detectable as a color change that is not intended to measure absolute thermal stability. From the figure, the color of PVC/Pb was changed rapidly from white to light yellow within 30 min, and then a gradual change in color from light yellow to yellow, turning brown within 160 min. The color of the PVC/CaZn could maintain its original white color for approximately 40 min and then experienced a gradual change in color from light yellow to yellow, turning to dark brown within 130 min. The PVC/OBS revealed that no color change was obtained up to 50 min, then the sample changed to yellow and turned dark brown within 140 min. It was found that the addition of the OBS stabilizer in the PVC showed better results in maintaining the initial color than the PVC/Pb and the PVC/CaZn. This behavior may confirm that the OBS can restrain the PVC degradation effectively during its processing. A similar behavior was also observed in the PVC/organic stabilizer, i.e., PVC/N-monomethyl-6-amino-thiouracil (MATU), showing better results in maintaining the initial color compared to the PVC/ZnSt_2_ [16]. In addition, the degradation behavior by observation of color change in the PVC/OBS was significantly related to the highest fusion, degradation times, and the lowest gelation speed, which resulted in slower degradation. These characteristics would cause a higher quality of the PVC/OBS at the end of the processing.

In principle, the thermal degradation process of PVC involves three major steps: (1) the initiation of dehydrochlorination, (2) the elimination of hydrochloride (HCl) and simultaneous formation of conjugated double bonds, and (3) termination of the dehydrochlorination process. The exact mechanism of the thermal degradation of PVC is still controversial, but it is generally accepted that the thermal degradation process with the evolution of HCl via chain reaction called zipper elimination or unzipping process takes place [1,17,18,19,20]. Color change of the PVC from the thermal degradation process has been attributed to the formation of a conjugated polyene sequence of 5 to 25 double bonds. Therefore, the incorporation of heat stabilizers such as Pb, CaZn, and OBS can stabilize or prolong its thermal integrity by preventing the chain reaction of degradation of the PVCs as a result of processing temperature and time. From Figure 2, it was observed that the stabilization of OBS and CaZn in the PVC is better than Pb by maintaining the initial color, while Pb can help in the long term thermal stabilization of the PVC within 200 min. PVC/CaZn can holds a better initial color compared to the PVC/Pb and may be an effective HCl scavenger of strongly basic carboxylates derived from calcium, having little or no Lewis acidity. Zinc exhibits stronger Lewis acid features that form covalent carboxylates. It cannot only scavenge HCl but can also substitute carboxylate for the allylic chlorine atoms and thus terminates the growth of the polyene sequences [21]. However, the CaZn are generally the least effective option in terms of thermal performance and relatively high cost. In the case of PVC/OBS, more effective maintenance of the initial color than the PVC/Pb was also observed. This behavior can be attributed to the major component of patented OBS chemistry based on uracil [22]. The uracil-based stabilizer can retard the degradation process by scavenging the HCl/Cl radical generated during processing and suppresses the crosslinking, which is more effective than Pb and CaZn stabilizers.

### 3.3. Thermal Degradation Behaviors of PVC Stabilized with Various Heat Stabilizers

#### 3.3.1. Thermogravimetric Analysis (TGA)

The thermal stability of rigid PVC/Pb, PVC/CaZn, and PVC/OBS samples can be determined by monitoring the weight change that occurs as a function of temperature at a constant heating rate. In this study, it was investigated by TGA. Derivative thermogravimetry (DTG) and TGA thermograms of the PVC/Pb, PVC/CaZn, and PVC/OBS samples compared with PVC samples are depicted in Figure 3. It was found that the addition of heat stabilizers could enhance the degradation temperature of PVC in the first stage, evaluated at a 5% weight loss of the PVC/Pb, PVC/CaZn, and PVC/OBS at 295 °C, 293 °C, and 297 °C, respectively, while that of PVC was 276 °C. It showed that the OBS as an organic component could better prevent the chain reaction of degradation of the PVC sample than the PVC stabilized with Pb and CaZn. This behavior was similar to PVC/eugenol (organic compound) that showed a longer thermal stability value (Ts) than PVC, PVC/DBLC, PVC/Ca-Zn soap, and PVC/OTM [23].

Furthermore, the char yield (*CR*) of the samples at 800 °C was also obtained. The Pb, CaZn, and OBS stabilizers can increase the *CR* of the PVC obtained at 16.9%, 19.1%, and 14.3%, respectively, compared with PVC at around 7.9%. The *CR* of the PVC stabilized with OBS was lower than those of Pb and CaZn, confirming that the addition of OBS into PVC reduced char formation. However, the PVC/OBS sample can be classified as a self-extinguishing category (*LOI* > 21) according to the van Krevelen and Hoftyzer equation [24], which is the same as the PVC/Pb and PVC/CaZn samples.

The *CR* can be applied as a decisive factor for estimating the limiting oxygen index (*LOI*) of the polymers based on the van Krevelen and Hoftyzer equation, as presented in Equation (8).
(8)LOI=17.5+0.4CR

The calculated *LOI* values derived from their *CR* of the PVC/Pb, PVC/CaZn, and PVC/OBS were 24.3, 25.1, and 23.2, respectively, while the *LOI* value of the PVC was approximately 20.7. The results revealed that the addition of Pb, CaZn, and OBS stabilizers could help enhance the fire resistance of the PVC, and adding OBS can maintain the fire resistance of the PVC. This suggests that the formation of a solid char barrier would block the available oxygen caused by the generation of combustion barriers and delay the flame propagation by adiabatic effect. In addition, in the case of OBS, the organic compounds would decompose in CO_2_ and water, reducing the *LOI* value. However, despite this, the PVC/OBS still maintains an adequate *LOI* level.

Moreover, the thermal degradation process of the PVC and all stabilized-PVCs, as can be seen in Figure 3b, showed two stages, i.e., 250–400 °C and 400–550 °C, respectively. The first stage is mainly attributed to dehydrochlorination by the removal of hydrogen chloride from the PVC main chain, thus forming a conjugated polyene structure. The second stage is mainly attributed to the fracture of conjugated polyene structures to form small molecular weight linear or cyclic hydrocarbons, and finally tar formation [25,26]. In addition, it was observed that there is a substage in the first stage of the PVC, indicating the existence of different events in this stage, suggesting a complex dehydrochlorination mechanism. While all stabilized-PVC samples still showed the substage with more overlapping, it is possible that the addition of Pb, CaZn, and OBS in the PVC showed a less complex dehydrochlorination mechanism of the PVC.

#### 3.3.2. Thermal Degradation Kinetics Analysis

The thermal stability effects of Pb, Ca/Zn, and OBS on PVC were compared by TGA at diverse heating rates, i.e., 5, 10, and 20 °C/min from 30 to 800 °C in a nitrogen atmosphere. Figure 4 show TGA and DTG thermograms of the PVC/Pb, PVC/CaZn, and PVC/OBS. The DTG thermograms of all PVC samples showed two thermal degradation stages, and indicated that chemical bonds of the second degradation stage were more difficult to break than the first degradation stage. As mentioned in item 3.3.1, the behaviors proved that the stability of the chemical bonds of C-Cl were lower than the C-C bonds. Therefore, when PVC is pyrolyzed, the chemical bonds of C-Cl would break first, while weight loss of PVC in the second stage was assumed as the breaking of the C-C bonds [27]. In two thermal degradation stages, the first stage is considered as the most important stage because the C-Cl bond in the PVC, having less energy than the C-C bond, dissociates first, resulting in the volatilization of the hydrogen chloride molecule.

Therefore, to study the thermal degradation kinetics of the PVC stabilized with Pb, CaZn, and OBS, the first stage of dehydrochlorination in the PVC/Pb, PVC/CaZn, and PVC/OBS was the focus of this work. The conversion (extend of reaction) at different temperatures of the first stage for each heating rate is plotted in Figure 5.

It can be seen that an average conversion of 68.3% for the PVC/Pb, 69.5% for the PVC/CaZn, and 67.0% for the PVC/OBS was resulted by the end of the first stage. Therefore, it can be summarized that the dehydrochlorination stage was approximately 68%, 70%, and 67% of the conversion of the total degradation process of the PVC stabilized with Pb, CaZn, and OBS, respectively. 

To calculate the kinetic parameter, according to the Kissinger method, the average activation energy of the first thermal degradation stage (*E*_a1_) was evaluated from the slopes of the plots in Figure 6 and ordered as follows: PVC/OBS (140 kJ/mol) > PVC/Pb (132 kJ/mol) > PVC/CaZn (110 kJ/mol). While, according to the FWO method, the *E*_a1_ at each conversion of the PVC/Pb, PVC/CaZn, and PVC/OBS determined from the slopes of the plots in Figure 7 is depicted in Figure 8. The average *E*_a1_ of the PVC samples from Figure 8 showed the same trend, which was calculated by the Kissinger method such as PVC/OBS (155 kJ/mol) > PVC/Pb (147 kJ/mol) > PVC/CaZn (129 kJ/mol). It should be noted that the *E*_a1_ averaged from the Kissinger and FWO methods for the PVC/OBS was 148 kJ/mol, indicating that much more energy was required for thermal degradation than the other two samples, i.e., approximately 140 kJ/mol for the PVC/Pb and 120 kJ/mol for the PVC/CaZn. The addition of the OBS in PVC might decrease the rate of the dehydrochlorination process; thus the PVC/OBS was more stable than the PVC/Pb and the PVC/CaZn when the temperature was lower than 400 °C.

Moreover, in Figure 8, the activation energy for the first substage of the dehydrochlorination of PVC/Pb and PVC/CaZn tended to decrease from 10% conversion until approximately 40% conversion and increasing thereafter as similarly reported in Ref. [25]. These characteristics confirmed that the energy required to break the C-Cl bond in the first substage of dehydrochlorination is high. As a result, the formed HCl acted as a catalyst in the degradation reaction, where the interaction of HCl with polyenes forms a chemical species whose reaction with the normal monomer unit gives new sites for initiation, allowing a decrease in the activation energy. After 40% conversion, or in the second substage of dehydrochlorination, the participation of chlorine atoms added to a large number of polyene chain sequences is reflected by an increase in the activation energy [28,29]. In the case of PVC/OBS, the activation energy was increased with an increase of conversion in the first substage of the dehydrochlorination. It is possible that the uracil in the OBS stabilizer can retard the degradation process by scavenging the HCl/Cl radical generated during processing, resulting in requiring the higher energy to break the C-Cl bond [2,16].

### 3.4. Effect of Repeated Processing Cycles on Gelation Behavior of PVC Stabilized with Various Heat Stabilizers

Achieving the optimum level of gelation or fusion can cause a variety of faults and failures in order to obtain the maximum mechanical properties. The gelation embraces changes involved in the conversion of separated particles of the polymer to a more or less continuous polymeric matrix. The morphology of PVC resin particles is complex, and it has been known for a long time. This can have a major effect on the processing and the mechanical properties of the finished product [20].

To find the degree of gelation, DSC thermograms of PVC/Pb, PVC/CaZn, and PVC/OBS compounds are plotted in Figure 9. The PVC compounds exhibited two endothermic peaks. The glass transition temperature of three PVC compounds measured at the first transition was approximately 90 °C. Then, a small additive peak was observed at 110 °C, 105 °C, and 115 °C for the PVC/Pb, PVC/CaZn, and PVC/OBS, respectively. Moreover, the degree of gelation (G) of PVC compounds stabilized with each heat stabilizer was estimated from the area of lower melting endotherm, which was divided by the sum of the areas of lower and higher melting endotherms and can be calculated using the following equation [30]:(9)$$G\left(\%\right)=\frac{H_{A}}{H_{A}+H_{B}}\times 100$$
where *G* is degree of gelation (%), *H_A_* is melting enthalpy of primary crystallites (J/g), and *H_B_* is the melting enthalpy of secondary crystallites (J/g).

From Figure 9, the percent of the degree of gelation of the PVC stabilized with Pb, CaZn, and OBS processed during the first by two-roll mills at 180 °C for 3 min was calculated to be 64.0%, 70.2%, and 59.0%, respectively. Types of heat stabilizers, therefore, showed significant effects on the degree of gelation of the starting PVC compounds. These characteristics may depend on the additive formulations in each heat stabilizer that resulted in different ease of processing or other necessary properties of the PVC compounds.

In addition, the degree of gelation determined from the DSC thermogram for each PVC compound as a function of repeated processing cycles is plotted in Figure 10. It was observed that the degree of gelation of each PVC compound from 1 to 5 processing cycles showed similar trends. An increase of the repeated processing cycle resulted in an increase in the degree of gelation up to the third cycle for the PVC/Pb and PVC/CaZn, and up to the fourth cycle for the PVC/OBS from the first processing cycle (or the original granules). It then tended to decrease in the fifth cycle. This was due to an increase in the number of repeated processing cycles which resulted in smaller grains (crystallites). There were remaining crystallite after each repeated processing cycle, being molten until up to the third or fourth cycle. A decrease in the degree of gelation was observed in the fifth cycle.

### 3.5. Mechanical Property as a Function of Repeated Processing Cycle of Rigid PVC Stabilized with Various Heat Stabilizers

A notched Izod impact test was used in studying an ability to absorb energy during plastic deformation, such as toughness and the ability of impact resistance [31]. The impact strength of the PVC stabilized with various types of heat stabilizers as a function of the number of processing cycles are plotted in Figure 11. It was found that the PVC/OBS showed a higher impact strength than the PVC/Pb and the PVC/CaZn, while the degree of gelation of the PVC/OBS was lower when compared to the above materials. It is possible that the more effective use of the OBS stabilizer one-packs provided the improved mechanical property of the PVC/OBS sample.

In addition, the impact strength of all PVC samples tended to increase from the first to the third processing cycle and then slightly decreased at the fourth processing cycle. It is possible that an increased processing cycle of up to a third can enhance a more continuous polymeric matrix which can be related to an increased degree of gelation, which in turn increases the impact strength. An increment of the degree of gelation in the fourth processing cycle tended to decrease the impact strength of all PVC samples. This indicated that the sample was changed to become more brittle.

The impact strength values of the PVC/Pb, PVC/CaZn, and PVC/OBS reprocessed up to five cycles were in the range of 102.7–104.8 J/m (10.3–10.5 kJ/m^2^), 103.8–106.3 J/m (10.4–10.6 kJ/m^2^), and 103.8–108.9 J/m (10.4–10.9 kJ/m^2^), respectively. It was found that the impact strength of the rigid PVC stabilized with Pb, CaZn, and OBS reached the requirement of commercial PVC pipes, i.e., 3.98 kJ/m^2^ and 5.0 kJ/m^2,^ as reported by Georg Fischer Harvel LLC and Thai Pipe Industry Co., Ltd., respectively [32,33]. However, the PVC/OBS still showed higher impact strength than the PVC/Pb and the PVC/CaZn. Therefore, the OBS can help to preserve the impact strength of the rigid PVC sample. In addition to the rather high toxicity of the Pb stabilizer and the least effective option in terms of thermal performance of CaZn stabilizer, it seems that the OBS can be stabilized to substitute both Pb and CaZn stabilizers.

### 3.6. Heat Distortion Temperature as a Function of Repeated Processing Cycle of Rigid PVC Stabilized with Various Heat Stabilizers

Heat distortion temperature (HDT) is the measure of a polymer’s resistance to distortion under a given load at elevated temperatures (short-term heat resistance) and is a useful indicator of the temperature limit above that region where the material cannot be used for structural applications. From Figure 12, the more eco-friendly heat stabilizers such as CaZn and OBS, compared to Pb stabilizer, can still maintain the HDT of the PVC at the first processing cycle at 76 °C for the PVC/OBS, 74.5 °C for the PVC/CaZn, and 75 °C for the PVC/Pb. Moreover, it was observed that the softening temperature of the rigid PVC stabilized with Pb, CaZn, and OBS reached the requirement of commercial PVC pipes, i.e., 70 °C and 75–82 °C, as reported by Georg Fischer Harvel LLC and Thai Pipe Industry Co., Ltd., respectively [32,33]. In addition, the effect of the repeated processing cycle of the rigid PVC samples on the HDT was also reported. It seems that the HDT of all PVC samples was relatively constant in each repeated processing cycle, and the PVC/OBS showed better heat resistance than the PVC/CaZn and the PVC/Pb within five repeated processing cycles. It is expected that the PVC/OBS can be used in wider applications that require heat resistance at higher temperatures.

## 4. Conclusions

Considering environmentally friendly heat stabilizers for rigid PVC pipes, PVC stabilized with the organic based stabilizer (OBS) was prepared and the rheological behavior, color change, and thermal stability were compared with PVC stabilized with Pb and CaZn. In addition, heat and impact resistance at repeated processing cycles of the PVC stabilized with Pb, CaZn, and OBS were also compared. The obtained PVC samples were characterized using a torque rheometer, DSC, UTM, impact tester, and TGA analyzer. As a result, the Pb stabilizer provided higher long-term heat stability rather than CaZn and OBS stabilizers in terms of the color change of the PVC, while OBS and CaZn can prolong the thermal stability time by maintaining the initial color better than Pb in the PVC. The increment of repeated processing up to four cycles was found to increase the degree of gelation and impact strength and showed no effect on the heat distortion temperature of all rigid PVC samples. The addition of Pb, CaZn, and OBS can significantly enhance the thermal stability, i.e., degradation temperature and char yield of the rigid PVC samples. The thermal degradation kinetics of the PVC/Pb, PVC/CaZn, and PVC/OBS samples were studied by a non-isothermal multiple scan rate method. It was found that there are two distinct stages in the thermal degradation process of the samples. The first stage for dehydrochlorination released 68.3%, 69.5%, 67.0% mass basis for the PVC/Pb, PVC/CaZn, and PVC/OBS, respectively. Two mathematical methods were used to determine the activation energy of the dehydrochlorination stage and found the following results: Kissinger’s method, *E_a_* for the PVC/Pb = 132 kJ/mol, *E_a_* for the PVC/CaZn = 110 kJ/mol, and *E_a_* for the PVC/OBS = 140 kJ/mol; and the Flynn–Wall–Ozawa method, *E_a_* for the PVC/Pb = 147 kJ/mol, *E_a_* for the PVC/CaZn = 129 kJ/mol, and *E_a_* for the PVC/OBS = 155 kJ/mol. Higher energy is required for the thermal degradation of PVC/OBS which decrease the rate of dehydrochlorination process in the PVC. Therefore, the organic based stabilizer showed high potential to be used as a heat stabilizer for rigid PVC for pipe applications due to good heat resistance and high mechanical property

## Figures and Tables

**Figure 1 polymers-14-00133-f001:**
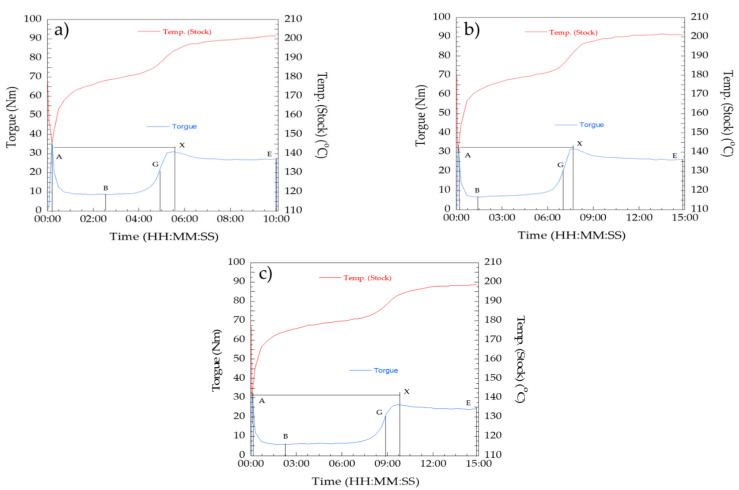
Plastograms of PVC compounds: (**a**) PVC/Pb, (**b**) PVC/CaZn, (**c**) PVC/OBS.

**Figure 2 polymers-14-00133-f002:**
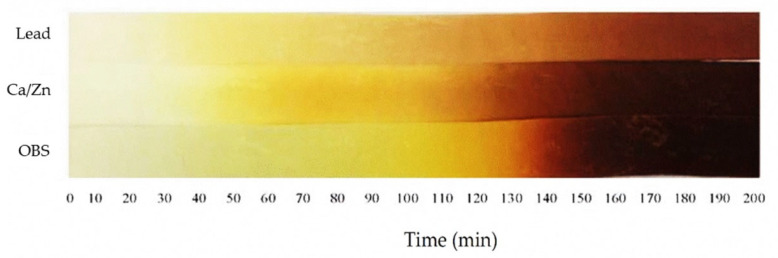
Color change of rigid PVC stabilized with various heat stabilizers. (All sample strips were processed by two-roll mills at 180 °C for 3 min and then automatically moved out of the box at 1 mm/min by a Mathis thermotester at 180 °C under air atmosphere).

**Figure 3 polymers-14-00133-f003:**
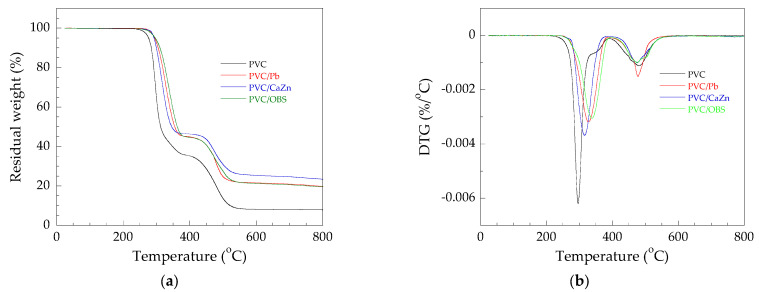
(**a**) TGA thermograms and (**b**) DTG thermograms of rigid PVC stabilized with various stabilizers.

**Figure 4 polymers-14-00133-f004:**
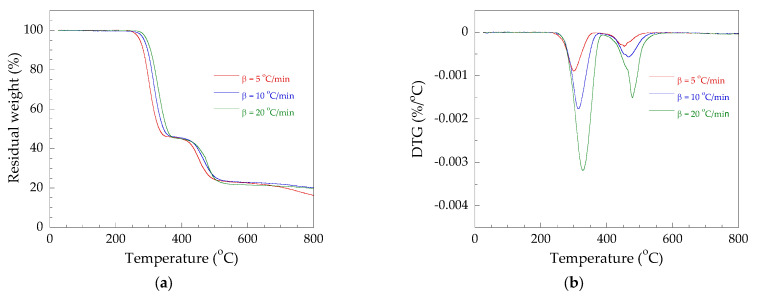
TGA and DTG thermograms of PVC/Pb (**a**,**b**), PVC/CaZn (**c**,**d**), PVC/OBS (**e**,**f**).

**Figure 5 polymers-14-00133-f005:**
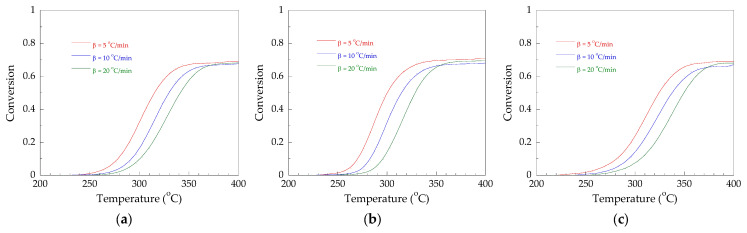
Conversion as a function of the temperature to dehydrochlorination stage: (**a**) PVC/Pb, (**b**) PVC/CaZn, (**c**) PVC/OBS.

**Figure 6 polymers-14-00133-f006:**
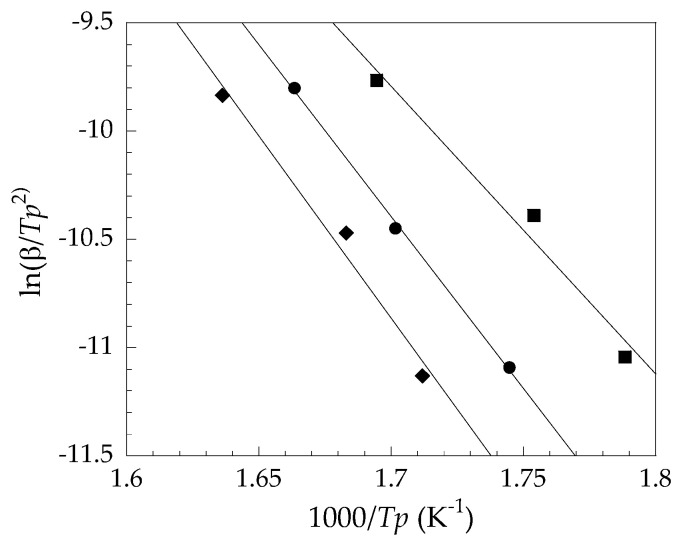
Kissinger method plots for average activation energy determination of the dehydrochlorination stage: (●) PVC/Pb, (■) PVC/CaZn, (◆) PVC/OBS.

**Figure 7 polymers-14-00133-f007:**
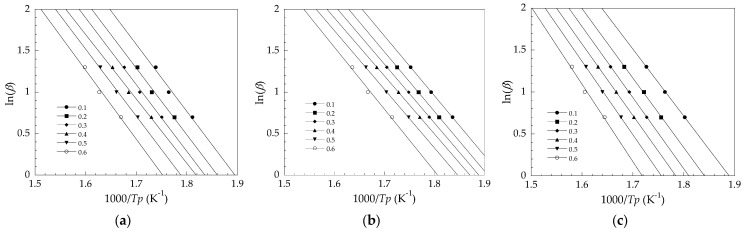
Flynn–Wall–Ozawa (FWO) plots for average activation energy determination of the dehydrochlorination stage: (**a**) (PVC/Pb, (**b**) (PVC/CaZn, (**c**) PVC/OBS.

**Figure 8 polymers-14-00133-f008:**
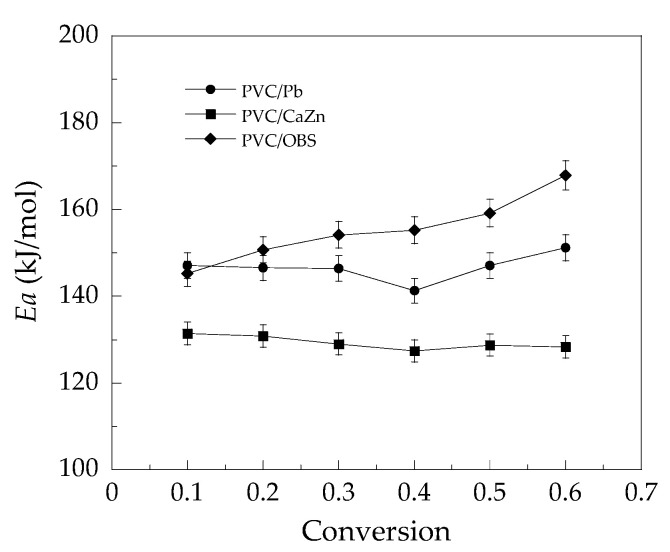
Activation energy as a function of conversion of the dehydrochlorination stage according to the Flynn–Wall–Ozawa (FWO) method.

**Figure 9 polymers-14-00133-f009:**
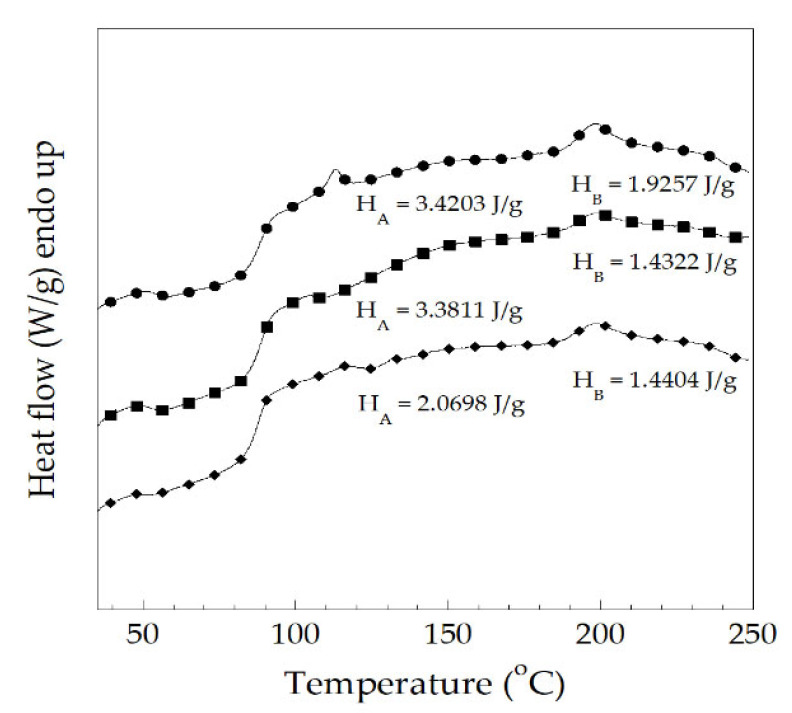
DSC thermograms of PVC stabilized with various types of heat stabilizers: (●) PVC/Pb, (■) PVC/CaZn, (◆) PVC/OBS. (all samples were processed by two-roll mills at 180 °C for 3 min during the first cycle).

**Figure 10 polymers-14-00133-f010:**
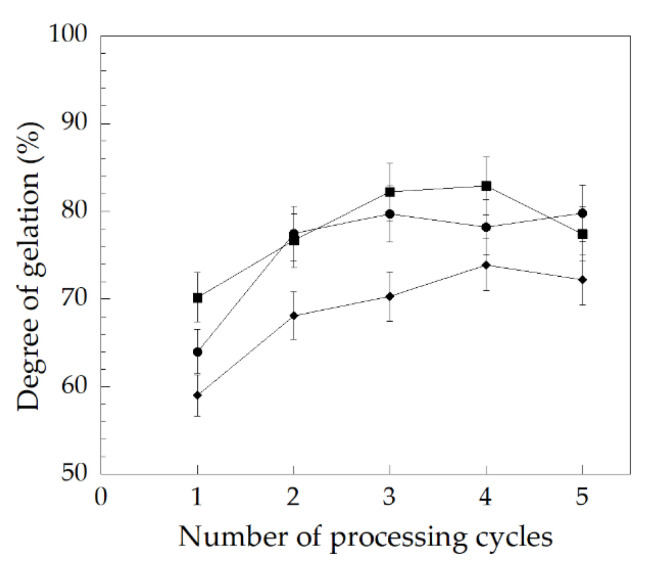
Effect of repeated processing cycle on the degree of gelation of PVC compound with various heat stabilizers: (●) PVC/Pb, (■) PVC/CaZn, (◆) PVC/OBS.

**Figure 11 polymers-14-00133-f011:**
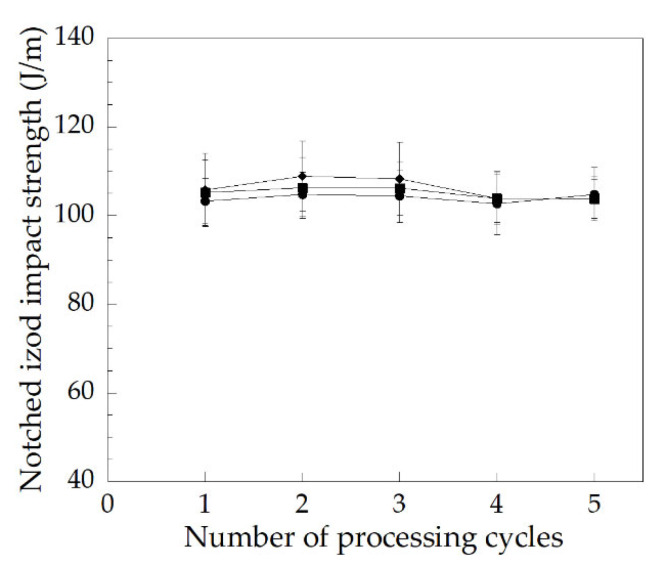
Relationship between repeated processing cycles and notched Izod impact strength of rigid PVC with various heat stabilizers: (●) PVC/Pb, (■) PVC/CaZn, (◆) PVC/OBS.

**Figure 12 polymers-14-00133-f012:**
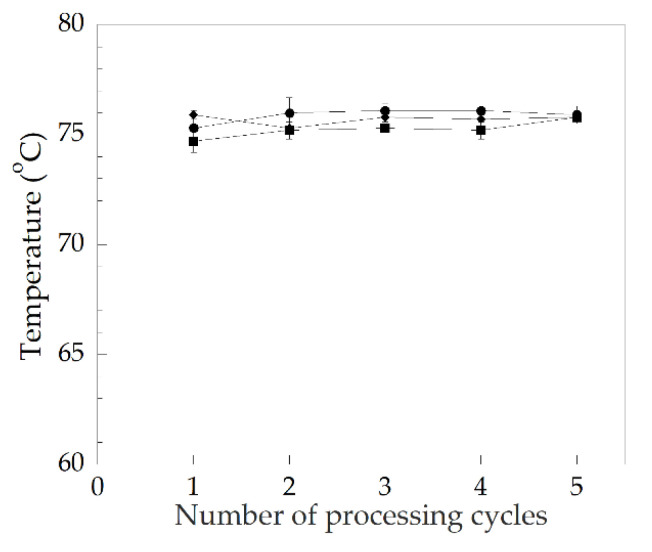
Effects of repeated processing cycles on heat distortion temperature of PVC with various heat stabilizers: (●) PVC/Pb, (■) PVC/CaZn, (◆) PVC/OBS.

**Table 1 polymers-14-00133-t001:** Maximum torque and fusion time of PVC compounds.

PVC Compound	Brabender Plasticorder Value	Torque (Nm)	Time (s)	Stock Temp. (°C)	Gelation Speed (Nm/min)	Fusion Time (A–X) (s)
PVC/Pb	Loading peak, A Minimum, B Inflection point, G Maximum, X End, E	40 8.5 19 31.0 27.5	10 135 290 330 600	150 178 187 193 202	5.64	320
PVC/CaZn	Loading peak, A Minimum, B Inflection point, G Maximum, X End, E	39 6.5 18.5 31.5 26.5	8 82 420 460 900	148.5 172.5 185 192 201	4.11	452
PVC/OBS	Loading peak, A Minimum, B Inflection point, G Maximum, X End, E	33.5 6.5 16.5 26.7 23.0	5 135 527 586 900	143.5 174 187 193 198.5	2.73	581

## Data Availability

Data is contained within the article.

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
