# Peer review of "Effects of Organic Based Heat Stabilizer on Properties of Polyvinyl Chloride for Pipe Applications: A Comparative Study with Pb and CaZn Systems"

_polymers, 2021, doi:10.3390/polym14010133_

Round 1
Reviewer 1 Report
The manuscript is well structured and the results and conclusions are well justified. The work itself contributes to building a fundamental understanding of OBS stabilizer(s), which could be helpful for PVC application and processing. A few comments are made below to help further improve the paper:
- Would be nice to polish the language, especially the abstract. Also, the sentence was incomplete at the end of Page 2, "The natural heterogeneity...systems.."
- What type of OBS was used in this work?
- Can you compare the performance of the OBS in this work with the ones reported in other literature? Recommending adding references to the performances of OBS stabilized PVC in the paper.
Author Response
Dear Reviewer,
Please find the attached file, the authors' response to review comment.
Best Regards,
Authors

Reviewer 2 Report
The manuscript is very well written, it is a very nice work. I suggest accept the manuscript directly. Just one comment, the authors in the text and in the conclusions use the term "mechanical properties", although in fact only the impact strength was measured. It need to be corrected.
Author Response

(The authors gave the same response as above.)
